# The cGAS/STING Pathway: Friend or Foe in Regulating Cardiomyopathy

**DOI:** 10.3390/cells14110778

**Published:** 2025-05-25

**Authors:** Weiyue Wang, Yuanxu Gao, Hyun Kyoung Lee, Albert Cheung-Hoi Yu, Markus Kipp, Hannes Kaddatz, Jiangshan Zhan

**Affiliations:** 1Institute for AI in Medicine, Faculty of Medicine, Macau University of Science and Technology, Macau 999087, China; 2Departments of Pediatrics, Baylor College of Medicine, Houston, TX 77030, USA; 3Jan and Dan Duncan Neurological Research Institute, Texas Children’s Hospital, Houston, TX 77030, USA; 4Neuroscience Research Institute, School of Basic Medical Sciences, Peking University Health Science Center, Beijing 100191, China; 5Institute of Anatomy, Rostock University Medical Center, Gertrudenstraße 9, 18057 Rostock, Germany; 6Department of Neurology, Rostock University Medical Center, Schillingallee 36, 18057 Rostock, Germany

**Keywords:** cardiomyopathy, cGAS/STING pathway, mitochondria, DNA damage

## Abstract

Inflammation is a central hallmark of cardiomyopathy, where misdirected immune responses contribute to chronic myocardial dysfunction. Among the emerging molecular mechanisms implicated in this process, the cyclic GMP–AMP synthase (cGAS)/stimulator of interferon genes (STING) signaling pathway has garnered increasing attention. Acting as a key cytosolic DNA sensor, the cGAS/STING pathway orchestrates inflammatory responses triggered by microbial infections or endogenous cellular stressors such as autophagy and apoptosis. Despite its pivotal role, the precise molecular mechanisms regulating this pathway and its role in cardiomyopathy-associated inflammation remain poorly understood and subject to ongoing debate. To address this scientific gap, we first reviewed key findings on cGAS/STING signaling in various forms of cardiomyopathy, drawing from in vivo and in vitro studies, as well as clinical samples. In the next step, we explored how the cGAS/STING pathway could be modulated by specific agonists and antagonists in the context of cardiac disease. Finally, by integrating publicly available human single-cell RNA sequencing (scRNA-seq) data and a systematic literature review, we identified existing molecular interventions and highlighted promising therapeutic targets aimed at mitigating cGAS/STING-driven inflammation. This comprehensive approach emphasizes the therapeutic potential of targeting the cGAS/STING pathway and provides a foundation for developing novel interventions aimed at alleviating inflammatory cardiomyopathy and improving patient outcomes. Future studies will be essential to validate these findings and facilitate their translation into clinical practice.

## 1. Introduction

Cardiomyopathy refers to a group of cardiac disorders characterized by structural and functional abnormalities of the heart muscle [1]. Cardiomyopathies can be divided into dilated cardiomyopathy (DCM), hypertrophic cardiomyopathy (HCM), restrictive cardiomyopathy, arrhythmogenic right ventricular cardiomyopathy, and other unclassified cardiomyopathies [2,3]. Clinical manifestations vary depending on the subtype and disease severity, but commonly include dyspnea, fatigue, chest pain, arrhythmias, and, in severe cases, sudden cardiac death [1]. Current therapeutic strategies primarily aim to alleviate symptoms and prevent complications. However, no specific pharmacological treatment has been approved that directly targets the underlying cause of cardiomyopathies [4]. This highlights the urgent need to elucidate the disease mechanisms to uncover novel therapeutic targets.

Numerous studies have explored signaling pathways in various cardiomyopathy models and revealed alterations in several key cellular and metabolic processes, including β-adrenergic signaling, MAPK/ERK signaling, WNT signaling, Hippo-Yes-associated protein signaling, CaM-kinase signaling, and autophagy signaling [5,6,7,8]. In recent years, the cyclic GMP–AMP synthase/stimulator of interferon genes (cGAS/STING) pathway has garnered increasing attention for its role in sensing cytosolic DNA and triggering downstream inflammatory responses and cell death [9,10,11,12]. Despite these advances, the specific involvement of the cGAS/STING pathway in cardiomyopathy, as well as its underlying regulatory mechanisms, remains poorly understood. This review seeks to provide a comprehensive overview of molecular interventions targeting the cGAS/STING pathway in the context of cardiomyopathy.

## 2. Overview of the cGAS/STING Signaling Pathway

The cGAS/STING pathway is a key component of the innate immune response which detects the presence of cytoplasmic DNA and subsequently triggers the expression of inflammatory genes. Cytosolic DNA can originate from both exogenous and endogenous sources [13]. While exogenous DNA typically originates from invading pathogens such as viruses and bacteria, endogenous DNA is often released during processes like the inefficient clearance of apoptotic cells, necrosis, or mitochondrial damage. As shown in Figure 1, the core components of this pathway include cyclic GMP–AMP synthase (cGAS), cGAMP, stimulator of interferon genes (STING), TANK-binding Kinase 1 (TBK1), interferon regulatory factor 3 (IRF3), type I Interferons (e.g., IFN-β), and nuclear factor kappa-light-chain-enhancer of activated B cells (NF-κB). cGAS is ubiquitously expressed in the cytoplasm and functions as a DNA sensor [14]. Upon binding to double-stranded DNA (dsDNA), it undergoes a conformational change and catalyzes the synthesis of cGAMP from Adenosine triphosphate (ATP) and Guanosine-5′-triphosphate (GTP). cGAMP then acts as a second messenger and activates STING. The activation of STING mediates its translocation from the endoplasmic reticulum membrane to the endoplasmic reticulum–Golgi intermediate compartment and Golgi apparatus, phosphorylating TBK1. Activated TBK1 subsequently phosphorylates and activates IRF3. Once phosphorylated, IRF3 translocates into the nucleus and induces the expression of type I interferons (e.g., IFN-β and IFN-α) [15,16]. In parallel, STING activation also stimulates the NF-κB signaling pathway, leading to the upregulation of pro-inflammatory cytokines.

## 3. Activation of the cGAS/STING Pathway in Different Cardiomyopathies

Inflammation plays a central role in the onset and progression of various forms of cardiomyopathy [17,18,19,20]. Recent evidence suggests that cGAS/STING-mediated inflammation and apoptosis are involved in myocardial injury and remodeling, potentially driving the development of cardiomyopathy [21,22,23]. To elucidate the critical role of cGAS/STING signaling in inflammation-mediated cardiomyopathies, we first systematically review the most important studies investigating this pathway across different cardiomyopathy subtypes (see Table 1).

### 3.1. Dilated Cardiomyopathy

Dilated cardiomyopathy (DCM) is characterized by ventricular enlargement and impaired systolic function of one or both ventricles. It is a major cause of heart failure and represents one of the leading indications for heart transplantation [42,43,44]. Up to 40% of all DCM cases in humans are of genetic origin, with most mutations affecting genes coding for cytoskeletal or contractile proteins [45,46]. The following section provides a systematic overview of the most prevalent genetic mutations associated with DCM, the corresponding animal models used to study them, and explores potential links between these mutations and dysregulation of the STING pathway.

#### 3.1.1. Lamin A/C (*LMNA*) Cardiomyopathy

Mutations in the *LMNA* gene (heterozygous loss-of-function or dominant-negative mutations), which encodes the nuclear envelope proteins lamin A and C, represent the second-most common genetic cause of DCM [47,48]. Lamin A/C are integral components of the nuclear lamina, providing structural support to the nucleus and serving as a platforms for protein interactions involved in gene regulation, DNA replication, and genome stability [49,50,51,52]. The disruption of lamin A/C compromises nuclear architecture, impairs DNA repair mechanisms, and alters gene expression profiles [53,54,55,56].

In 2022, Cheedipudi et al. demonstrated that the DNA damage response pathway is activated in *Lmna*-deficient mice. They also found that genetically inhibiting cGAS in these mice improved survival, enhanced cardiac function, and reduced tissue fibrosis and programmed cell death [29]. They observed the robust upregulation of several components of the cGAS/STING pathway in the heart of *Lmna*-deficient mice, including cGAS, TBK1, STING1, pIRF3, and NF-kB. The genetic knockout of cGAS in these mice significantly attenuated all these factors except for STING1 [29]. However, in 2024, En et al. challenged these findings and proposed that cGAS/STING does not contribute to LMNA cardiomyopathy in adult mice. They found that the deletion of cGAS or STING failed to rescue the cardiac phenotype in adult *Lmna*-deficient mice [30]. They reported that cGAS/STING was not activated in cardiomyocytes and attributed this to the generally low expression levels of cGAS and STING in adult cardiac myocytes. Instead, through single-nucleus RNA-sequencing (snRNA-seq) and gene enrichment analyses, they proposed that extracellular matrix (ECM) signaling—not cGAS/STING—is the dominant inflammatory mediator in this context [30]. More recently, Zuela-Sopilniak et al. performed a multi-layered transcriptomic analysis of LMNA-related DCM and identified two cardiomyocyte subclusters implicated in disease pathogenesis [57]. Despite low cGAS expression, STING activation was observed—potentially driven by an alternative DNA sensor, interferon-gamma-inducible protein 16 (IFI16). The authors hypothesized that cGAS-independent cytosolic DNA sensing and DNA damage response pathways in cardiomyocytes may initiate transcriptional reprogramming in cardiac fibroblasts, promote immune cell recruitment, and activate ECM remodeling, ultimately leading to cardiac dysfunction [57].

Several factors may account for the conflicting findings between these studies. First, the LMNA-related DCM animal models employed by the respective research groups differed significantly. Cheedipudi et al. used *Myh6-Cre:Lmna*^F/F^ mice in which LMNA was deleted postnatally, whereas En et al. used an inducible Cre recombinase system to delete LMNA specifically in cardiomyocytes at 6–8 weeks of age [30]. The timing of LMNA deletion may critically influence the activation of downstream signaling pathways, particularly given the dynamic changes that occur during cardiac development. It remains unclear which model more accurately recapitulates the human form of LMNA-related cardiomyopathy. Second, the studies differed in their methods for assessing cGAS/STING pathway activity. Cheedipudi et al. employed immunoblot analysis to assess the expression levels of key proteins involved in the cGAS/STING pathway, while En et al. investigated the activity of key genes from this pathway at the transcriptional level. However, mRNA and protein expression levels often correlate poorly due to post-transcriptional regulation and post-translational modifications. Thus, transcript abundance may not reliably reflect functional pathway activity. This methodological discrepancy may explain the divergent findings reported by En et al. and Zuela-Sopilniak et al. compared to those of Cheedipudi et al. [29,30,57].

#### 3.1.2. LEM Domain-Containing Protein 2 (LEMD2)-Associated Cardiomyopathy

Similar to lamin A/C, the LEM domain-containing protein 2 (LEMD2) is a critical component of the nuclear envelope, contributing to nuclear structure and integrity [58,59]. Mutations in the *LEMD2* gene, most notably the c.T38 > G (p.L13R) variant in humans, are associated with a form of dilated cardiomyopathy that often presents with arrhythmic features [60,61,62].

In 2022, Caravia et al. generated the first LEMD2-associated cardiomyopathy mouse model to investigate the role of LEMD2 in cardiac development and disease. Their findings revealed the activation of both the DNA damage response and apoptotic pathways in LEMD2 cardiac-specific knockout hearts. Immunofluorescence staining of heart tissue and isolated cardiomyocytes showed elevated levels of phosphorylated histone H2AX (γ-H2AX), confirming the presence of DNA damage [61]. In 2023, Chen et al. demonstrated that in LEMD2 mutant mouse cardiomyocytes and HeLa cells, persistent nuclear envelope rupture resulted in the cytoplasmic leakage of DNA repair factors, triggering cell cycle arrest. Additionally, the rupture of the nuclear envelope exposed genomic DNA to the cytosol, activating the cGAS/STING interferon (IFN) signaling pathway. The activation of this pathway promoted the expression of senescence-associated secretory phenotype (SASP) factors, thereby driving cellular senescence [40].

### 3.2. Diabetic Cardiomyopathy

Diabetic cardiomyopathy is a distinct cardiac condition that arises from diabetes-induced alterations in myocardial structure and function, independent of other cardiovascular risk factors such as hypertension or coronary artery disease. It is characterized by myocardial hypertrophy, inflammation, fibrosis, and cardiomyocyte death [63,64]. If left untreated, diabetic cardiomyopathy can progress to heart failure [65]. Unlike other forms of cardiomyopathy, diabetic cardiomyopathy is uniquely tied to metabolic disturbances, including increased fatty acid utilization and oxidative stress, which contribute to its pathogenesis and complicate treatment [63,66,67].

In 2022, Ma et al. reported the activation of the cGAS/STING pathway in an obesity-related DCM mouse model, driven by increased cytosolic mitochondrial DNA (mtDNA) [31]. Transmission electron microscopy revealed significant alterations in mitochondrial structure, and elevated levels of free cytosolic mtDNA were confirmed using co-immunolabeling and qRT-PCR. Concurrently, the upregulation of cGAS, STING, and downstream effectors such as NF-κB, IRF3, and IL-1β was observed at both the mRNA and protein levels in the hearts of diabetic mice. To demonstrate the causal role of mtDNA in activating cGAS, purified mtDNA was transfected into H9C2 cells (cardiac muscle cell line), which resulted in increased expressions of cGAS, STING, IL-1β, and IL-18, along with STING aggregation at the Golgi apparatus, indicating its functional activation. Both the genetic and pharmacological inhibition of STING improved cardiac function, reduced myocardial hypertrophy and fibrosis, and attenuated inflammation in this mouse model of diabetic cardiomyopathy [31]. Around the same time, another study highlighted that oxidative mitochondrial damage caused by lipid toxicity in diabetic cardiomyopathy led to the cytosolic release of mtDNA, subsequently activating the cGAS/STING pathway. In addition, they showed that NLR family pyrin domain containing 3 (NLRP3) inflammasome-dependent pyroptosis and pro-inflammatory responses were also activated as downstream pathways, contributing to myocardial hypertrophy [32]. Building on these mechanistic insights, several studies have proposed therapeutic strategies targeting the cGAS/STING pathway in diabetic cardiomyopathy. In 2022, Lu et al. demonstrated that the cardiomyocyte-specific overexpression of Meteorin-like hormone (METRNL) ameliorated diabetic cardiomyopathy by simultaneously activating autophagy and inhibiting cGAS/STING signaling. Mechanistically, METRNL-induced Unc-51 Like Autophagy Activating Kinase 1 (ULK1) phosphorylation facilitated STING dephosphorylation and its mitochondrial translocation. There, STING formed a complex with tumor necrosis factor receptor-associated factor 2 (TRAF2), promoting its ubiquitination and degradation, thereby sensitizing cardiomyocytes to autophagy activation [33]. In 2023, another group showed that irisin—a myokine—rescued cardiac dysfunction in diabetic cardiomyopathy by activating mitochondrial ubiquitin ligase MITOL (also known as MARCH5) and suppressing NLRP3 inflammasome activity through the inhibition of the cGAS/STING pathway [36]. In 2024, Chen et al. revealed that deficiency of the chromatin remodeler Brahma-related gene 1 (BRG1/SMARCA4) led to the accumulation of cytosolic dsDNA and the hyperactivation of cGAS/STING signaling. This exacerbated inflammation and apoptosis in cardiomyocytes under hyperglycemic and hyperlipidemic conditions [34]. Simultaneously, Huang et al. demonstrated that the administration of recombinant IL-37 or induction of endogenous IL-37 expression alleviated cardiac dysfunction and fibrosis in diabetic cardiomyopathy mice [35]. They proposed a novel mechanism wherein hyperglycemia-induced mitochondrial damage—mediated by the SIRT1/AMPK/PGC1α axis—caused the release of mtDNA-containing extracellular vesicles. These vesicles were taken up by cardiac fibroblasts, activating Toll-like receptor 9 (TLR9) and cGAS/STING signaling, thereby initiating pro-fibrotic responses and adverse cardiac remodeling. IL-37 exerted its therapeutic effects by suppressing these pathways. Collectively, these findings highlight the central roles of mitochondrial dysfunction and cGAS/STING activation in the pathogenesis of diabetic cardiomyopathy. Targeting this pathway offers a promising therapeutic strategy to mitigate inflammation, fibrosis, and cardiac dysfunction associated with the disease.

### 3.3. Arrhythmogenic Cardiomyopathy

Arrhythmogenic cardiomyopathy (ACM), also known as arrhythmogenic right ventricular cardiomyopathy/dysplasia (ARVC/D), encompasses a group of inherited myocardial disorders characterized by ventricular arrhythmias, progressive heart failure, and an elevated risk of sudden cardiac death [68]. The majority of ACM cases are attributed to genetic mutations affecting desmosomal proteins, such as Desmoplakin (DSP), Plakophilin-2 (PKP2), Desmocollin-2 (DSC2), Desmoglein-2 (DSG2), and Junction Plakoglobin (JUP). In addition, mutations in non-desmosomal genes such as phospholamban (PLN), transmembrane protein 43 (TMEM43), and cadherin-2 (CDH2) have also been implicated in disease pathogenesis [69,70,71,72].

TMEM43 is a highly conserved transmembrane protein localized to the inner nuclear membrane and is thought to play a critical role in maintaining nuclear envelope integrity [73,74]. In 2021, Rouhi et al. found that a haploinsufficiency of TMEM43 activates the DNA damage response and the TP53 signaling pathway, resulting in increased expressions of senescence-associated secretory phenotype (SASP) factors and pro-fibrotic mediators in cardiomyopathy. Interestingly, the activation of cGAS and STING1—key components of the DNA damage response—was observed only at later disease stages, aligning with the delayed onset of cardiac symptoms in this model [37].

### 3.4. Doxorubicin-Induced Cardiomyopathy

Doxorubicin, an anthracycline antibiotic, is widely used in the treatment of various malignancies, including breast cancer, lymphomas, and leukemias [75]. Its antitumor effects are primarily mediated through DNA intercalation, the generation of reactive oxygen species (ROS), and the induction of apoptosis in rapidly dividing cancer cells [76]. However, the clinical use of doxorubicin is significantly limited by its well-documented cardiotoxicity [77,78,79]. Doxorubicin-induced cardiomyopathy (DIC) shares morphological and functional features with dilated cardiomyopathy and represents a serious, often irreversible complication. Current hypotheses regarding the pathogenesis of DIC include oxidative stress, mitochondrial dysfunction, DNA damage, calcium dysregulation, and chronic inflammation [80]. Dexrazoxane, an iron-chelating agent that reduces ROS formation, remains the only approved pharmacological agent for the prevention of DIC [81,82,83]. Recent studies have implicated the cGAS/STING pathway as a potential contributor to DIC pathogenesis through inflammation-driven mechanisms [21,22,41].

DIC can manifest in the following two distinct forms: acute and chronic. Acute DIC typically develops within hours to weeks following doxorubicin administration and is primarily driven by direct myocardial toxicity, oxidative stress, and acute inflammatory responses [84]. Chronic DIC, in contrast, may appear months or even years later and is characterized by persistent oxidative stress, mitochondrial dysfunction, ongoing DNA damage, and myocardial fibrosis [77]. In 2023, Xiao et al. demonstrated the significant upregulation of cGAS and STING protein levels in an acute DIC mouse model. The knockdown of STING extended survival, improved cardiac function, and mitigated structural cardiac damage, as evidenced by reduced myofibrillar vacuolation and preserved myofibril content. STING inhibition also attenuated cardiomyocyte apoptosis and inflammation, indicated by decreased cleaved-Caspase 3/Caspase 3 and BAX/BCL2 ratios, fewer TUNEL-positive cells, and reduced levels of pro-inflammatory cytokines [41]. In a chronic DIC mouse model, Luo et al. further confirmed cGAS/STING pathway activation in the heart and demonstrated that the global deletion of cGAS or STING prevented the development of DIC. Notably, they observed that cGAS/STING pathway activation occurred predominantly in cardiac endothelial cells and macrophages, rather than in cardiomyocytes or fibroblasts. Endothelial-cell-specific STING knockdown improved both cardiac function and endothelial integrity. Mechanistically, cGAS/STING activation in endothelial cells triggered inflammation and mitochondrial dysfunction via the CD38-mediated depletion of nicotinamide adenine dinucleotide (NAD⁺). Furthermore, this endothelial-specific signaling cascade also impaired cardiomyocyte mitochondrial bioenergetics by reducing NAD⁺ availability through CD38 ecto-NADase activity [21]. These findings highlight the pivotal role of cGAS/STING signaling in both acute and chronic forms of doxorubicin-induced cardiomyopathy, particularly through its effects on endothelial inflammation and mitochondrial dysfunction. Targeting this pathway may offer a promising therapeutic strategy to prevent or mitigate cardiotoxicity in patients undergoing anthracycline-based chemotherapy.

### 3.5. Sepsis-Induced Cardiomyopathy

Sepsis is a life-threatening condition characterized by a dysregulated systemic inflammatory response to infection [85,86]. If uncontrolled, this inflammatory cascade can lead to multiple organ dysfunction, including cardiovascular impairment [87]. Sepsis-induced cardiomyopathy (SIC) refers to the transient myocardial dysfunction that occurs during sepsis. Unlike many other forms of cardiomyopathy, SIC is typically reversible with timely resolution of the underlying infection and appropriate clinical management [88,89].

In 2019, Li et al. were the first to demonstrate that the STING–IRF3 axis contributes to lipopolysaccharide (LPS)-induced cardiac dysfunction, inflammation, apoptosis, and pyroptosis in SIC [24]. They showed that STING deficiency ameliorated these pathological responses and improved cardiac function and survival. Mechanistically, LPS did not affect total STING protein levels, but promoted its perinuclear translocation, as well as IRF3 nuclear translocation. STING knockdown inhibited IRF3 phosphorylation and nuclear localization, thereby attenuating IRF3-mediated pathological effects [24]. In contrast, Kong et al. reported that LPS administration led to an upregulation of STING protein expression and its activation. They identified islet cell autoantigen 69 (ICA69) as a novel positive regulator of STING in the context of septic cardiac injury. ICA69 and STING were found to colocalize in cardiac tissue and macrophages, and *Ica69* knockdown significantly reduced STING-driven inflammation and ferroptosis, thereby mitigating LPS-induced cardiac damage [25]. Further supporting the role of this pathway, in 2023, Liu et al. observed increased expressions of both cGAS and STING in the hearts of LPS-treated mice [26]. In normal H9C2 cardiomyocytes, the silencing of *cGAS* had no significant impact on downstream signaling. However, in LPS-stimulated cells, *cGAS* knockdown significantly reduced the expressions of STING, IRF3, and TBK1. Additionally, the suppression of *cGAS* alleviated LPS-induced inflammation, apoptosis, and excessive ROS production. Collectively, these findings underscore the critical involvement of the cGAS/STING signaling pathway in the pathogenesis of sepsis-induced cardiomyopathy, highlighting its potential as a therapeutic target in septic cardiac dysfunction [26].

### 3.6. Other Cardiomyopathies

Carnitine acetyltransferase (CRAT) is a mitochondrial enzyme responsible for converting acetyl-CoA to acetylcarnitine, facilitating the transport of acetyl groups out of the mitochondria [90]. CRAT is important in regulating cellular energy metabolism, and its deficiency results in mitochondrial dysfunction [91,92]. In a fibroblast-specific CRAT silencing model, mtDNA was found to be released into the cytosol, activating downstream cGAS/STING/NF-κB signaling pathways [93]. Expanding on these findings, in 2023, Mao et al. generated a cardiomyocyte-specific *Crat*-deficient mouse model which developed a dilated cardiomyopathy phenotype. They showed that the depletion of *Crat* promoted the release of mtDNA into the cytoplasm via the mitochondrial permeability transition pore (mPTP), triggering a type I interferon response within cardiomyocytes. Multiple cytosolic RNA and DNA sensors, including cGAS, Ddx58, Ifih1, and Absent In Melanoma 2 (AIM2), were activated. Importantly, the knockdown of *cGAS* reduced the expression of interferon-stimulated genes’ (ISGs) expression, inhibited AIM2 inflammasome activation, and improved cardiac contractile function in CRAT-deficient mice [27].

Chagas cardiomyopathy is a chronic cardiac manifestation of *Trypanosoma cruzi* infection, characterized by intense myocardial inflammation, fibrosis, and life-threatening arrhythmias [94,95,96]. In this context, Choudhuri et al. demonstrated that extracellular vesicles derived from *T. cruzi*-infected cells induced elevated levels of IL-1β, IL-6, and TNF-α in macrophages. The pharmacological inhibition of cGAS using PF-06928215 significantly reduced the production of these pro-inflammatory cytokines, implicating the cGAS/STING pathway in the inflammatory pathogenesis of Chagas cardiomyopathy [38].

Stress cardiomyopathy—also known as Takotsubo cardiomyopathy or “broken heart syndrome”—is an acute, reversible cardiac condition triggered by extreme emotional or physical stress. A surge in catecholamines such as adrenaline and norepinephrine is believed to play a central role in disease onset. Inflammation induced by this catecholamine surge has been shown to exacerbate myocardial injury and disease progression [97,98,99,100]. In 2024, in a stress cardiomyopathy mouse model, Wang et al. showed that acute catecholamine exposure induced necrotic death in cardiomyocytes, resulting in the release of self-DNA and other damage-associated molecular patterns (DAMPs) [39]. These DAMPs were detected by macrophages, activating cytosolic DNA-sensing pathways—most notably the STING axis—which triggered the production of pro-inflammatory cytokines, including Tumor necrosis factor (TNF), IL-6, and CCL2, thereby promoting myocardial inflammation and injury [39]. Interestingly, the study revealed that DNA sensing was not limited to cGAS; transcriptomic analysis identified the upregulation of several additional DNA sensors, including Ddx41, Ddx58, and Zbp1, suggesting a broader network of innate immune activation in stress cardiomyopathy.

## 4. Molecular Interventions and Potential Targets Within the cGAS/STING Pathway in Cardiomyopathy

### 4.1. Molecular Intervention in the cGAS/STING Pathway in Cardiomyopathy

Over the past decades, various animal models have been developed to explore the pathophysiological mechanisms underlying cardiomyopathy, significantly advancing our understanding of the role played by the cGAS/STING signaling pathway in disease progression. A comprehensive literature review revealed that the cGAS/STING pathway is consistently activated in multiple forms of cardiomyopathy—including dilated (DCM) and hypertrophic cardiomyopathy (HCM)—when compared to non-failing (NF) control hearts. Importantly, the inhibition of this pathway has been shown to markedly slow disease progression and improve cardiac outcomes across several experimental models.

Both genetic and pharmacological interventions targeting cGAS and STING have demonstrated therapeutic potential. Notably, the pharmacological inhibition of this pathway has emerged as a promising therapeutic approach. For example, RU.521, a selective cGAS inhibitor, has been shown to reduce myocardial fibrosis and improve cardiac function in DCM models [20]. Likewise, H-151, a covalent inhibitor of STING palmitoylation, suppresses type I interferon signaling and mitigates cardiac hypertrophy in pressure-overload-induced HCM [30]. Other STING inhibitors, such as C-176 and C-178, have exhibited robust anti-inflammatory effects in models of autoimmune myocarditis and ischemia–reperfusion injury [25,26,27,28]. Taken together, these findings underscore the therapeutic promise of targeting the cGAS/STING pathway in cardiomyopathy. A detailed summary of these pharmacological compounds, their mechanisms of action, and associated experimental models is presented chronologically in Table 2.

### 4.2. Mitochondrial Alteration as a Hotspot for cGAS/STING Pathway Activation

As illustrated in Figure 1, mitochondrial damage is a major trigger for the release of endogenous DNA into the cytosol, subsequently activating the cGAS/STING signaling pathway. Analysis of publicly available human single-cell RNA sequencing data from Chaffin et al. [108] revealed a significant reduction in mitochondrial transcript abundance in cardiomyocytes from individuals with cardiomyopathy (DCM and HCM) compared to non-failing (NF) controls (see Figure 2). These findings are consistent with recent work by Wang et al., who reported the marked downregulation of mitochondrial-function-related pathways—including the mitochondrial matrix, inner membrane, respiratory chain complex I, oxidative phosphorylation, and the electron transport chain—in a stress cardiomyopathy mouse model [39].

Cardiomyocytes rely heavily on mitochondrial oxidative phosphorylation to meet their high energetic demands. Several mitochondrially encoded genes, such as *MT-ND1*, *ND2*, *ND3*, *ND4*, *ND4L*, *ND5*, *ND6*, *CO1*, *CO2*, *CO3*, *ATP6*, and *CYB*, encode essential components of respiratory chain complexes I, III, IV, and V, all localized to the inner mitochondrial membrane. NADH dehydrogenase (*ND*) genes form the core of complex I, initiating electron transport and contributing to the proton gradient essential for ATP synthesis. *CYB* encodes cytochrome *b*, a key component of complex III, while *CO1*, *CO2*, and *CO3* encode for the catalytic subunits of complex IV. *ATP6*, on the other hand, forms part of the proton channel of ATP synthase (complex V) [109,110].

The dysfunction of these genes has been directly linked to cardiac pathologies. Mutations in *MT-ND1* and *MT-ND5* are associated with hypertrophic cardiomyopathy, while defects in *CYB*, *CO2*, and *CO3* are implicated in both hypertrophic and dilated cardiomyopathies [111,112,113]. The m.8993T > G mutation in *ATP6*, for instance, leads to ATP synthase deficiency and is frequently observed in early-onset cardiomyopathy with the Leigh or neuropathy-ataxia-retinitis pigmentosa (NARP) syndromes [114]. Beyond inherited mitochondrial disorders, reduced expressions of mitochondrial genes have also been reported in failing human hearts, underscoring the broader relevance of mitochondrial dysfunction in acquired cardiomyopathies [115,116].

These findings highlight the central role of mitochondrial bioenergetics in maintaining cardiac function and demonstrate how disruptions in mitochondrial gene expression can drive cardiomyopathy through impaired ATP production and increased oxidative stress. Furthermore, they suggest that mitochondrial instability facilitates cytosolic mtDNA release, potentially activating the cGAS/STING pathway and perpetuating harmful type I interferon responses. These pilot studies shed light on the importance of mitochondrial balance in the pathogenesis of cardiomyopathies.

Despite growing evidence, the specific DNA sensors and mechanisms responsible for recognizing mitochondrial DNA in the context of cardiomyopathy remain to be fully elucidated. Targeting mitochondrial dysfunction represents a promising therapeutic strategy—not only to restore mitochondrial homeostasis, but also to prevent aberrant activation of the cGAS/STING pathway and its downstream inflammatory consequences.

### 4.3. Future Perspectives in Understanding the cGAS/STING Pathway in Cardiomyopathy

Since Chen’s group at the University of Texas Southwestern Medical Center discovered the cGAS/STING pathway, cGAS has been recognized as a key cytosolic DNA-sensing mechanism that drives type I interferon production and innate immune responses [9,10]. Despite growing evidence implicating this pathway in cardiovascular disease, its precise role in the pathogenesis of cardiomyopathy remains incompletely understood. Several critical questions should be addressed to advance our understanding, as follows:(i)Does DNA damage directly contribute to disease progression, or is it merely a byproduct of late-stage damage to cardiomyocytes and progressive cell loss?(ii)How is the cGAS/STING pathway activated and regulated in myocytes compared to non-myocyte populations during cardiomyopathy (e.g., macrophages, fibroblasts, and endothelial cells)? What are the specific downstream targets of cGAS in degenerating cardiac muscle cells?(iii)Given that mtDNA has been shown to activate cGAS/STING signaling during cardiomyopathy [117], it is crucial to elucidate the mechanisms governing mtDNA release into the cytosol. Furthermore, can cytosolic mtDNA be selectively targeted as a therapeutic strategy to mitigate cardiac dysfunction?

Equally important is the consideration of the cGAS/STING pathway within the broader context of aging and cellular senescence. Aging is a well-established, independent risk factor for cardiomyopathy [118], and elderly individuals are disproportionately affected by cardiovascular disease. This demographic often experiences comorbid conditions, including neurodegenerative disorders, which collectively exacerbate disease severity, functional decline, and disability progression [119]. Given the known links between the cGAS/STING pathway, DNA damage, and senescence, future studies should explore whether this signaling axis acts as a central regulator of chronic cardiomyopathy, particularly in the aging heart. A deeper mechanistic understanding of these processes will be critical for identifying novel biomarkers and therapeutic targets, ultimately improving clinical outcomes for patients with cardiomyopathy.

## 5. Conclusions

Inflammation is a key factor in the development and progression of cardiomyopathy, and the cGAS/STING pathway has emerged as a significant contributor to this process. As a central sensor of cytosolic DNA, this pathway responds to both external pathogens and internal cellular stress, promoting immune activation. While evidence supports its involvement in inflammatory cardiomyopathy, the underlying mechanisms remain unclear. This review summarized current insights from experimental models and clinical studies, examined the effects of pathway modulators, and identified therapeutic targets using single-cell RNA sequencing data. Targeting cGAS/STING may offer promising new strategies to reduce inflammation and improve outcomes in cardiomyopathy.

## Figures and Tables

**Figure 1 cells-14-00778-f001:**
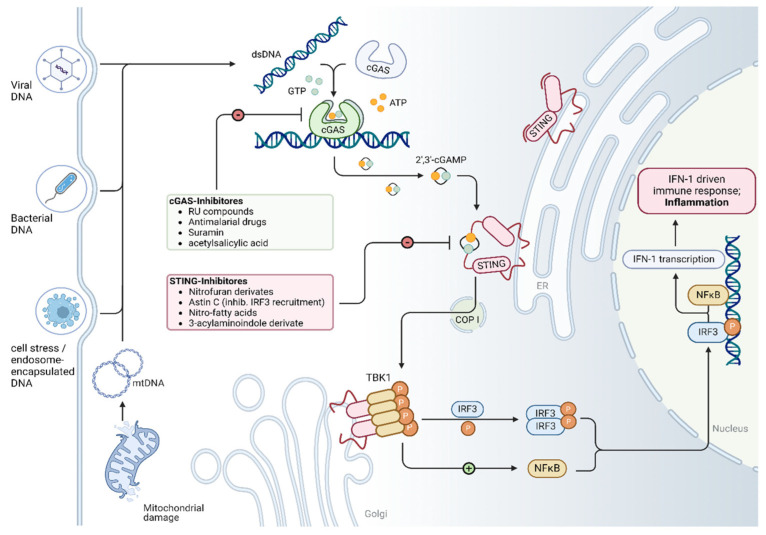
Schematic illustration of the cGAS/STING pathway. Cyclic guanosine monophosphate adenosine monophosphate synthase (cGAS), which is freely present in the cytosol, binds to both exogenous and endogenous double-stranded DNA (dsDNA). Viruses and bacteria can be sources of exogenous dsDNA, while endogenous dsDNA mainly originates from cell apoptosis, auto cellular death, and mitochondrial damage. After binding to dsDNA, cGAS catalyzes a reaction between ATP and GTP to form 2′,3′-cGAMP. 2′,3′-cGAMP then binds to the ‘stimulator of interferon genes’ (STING), located on the membrane of the endoplasmic reticulum (ER), triggers a conformational change and subsequently activates STING. The activated STING is transported to the Golgi apparatus via the coatomer protein complex I (COP I), where it phosphorylates TANK-binding kinase 1 (TBK1) to form a tetramer. TBK1 then phosphorylates interferon regulatory factor 3 (IRF3), which dimerizes and enters the nucleus together with nuclear factor kappa B (NF-κB) to induce the expression of type 1 interferons, as well as other proinflammatory cytokines. Many chemical compounds are known to modulate the cGAS/STING signaling pathway (see green and red box). Created in BioRender. Kaddatz, H. (2025) https://BioRender.com/9kzlnt0 (accessed on 17 May 2025).

**Figure 2 cells-14-00778-f002:**
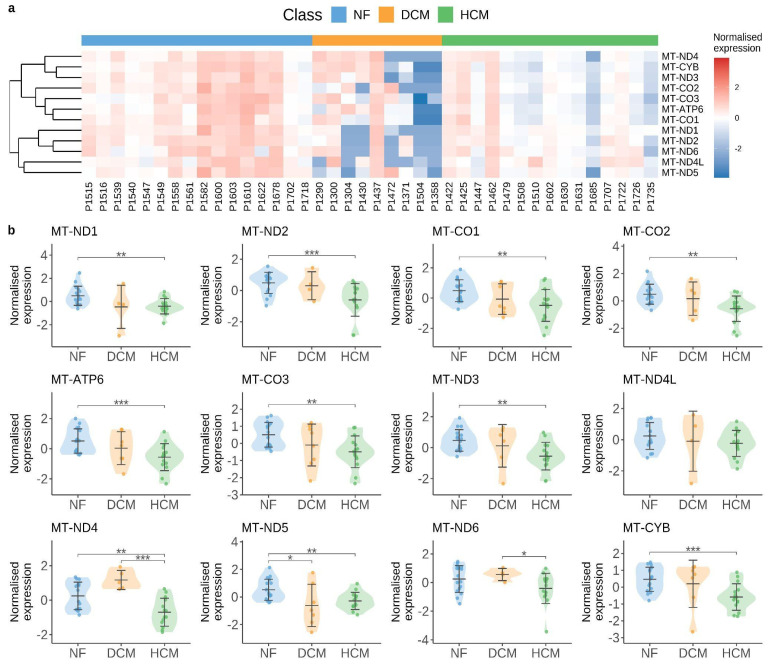
Mitochondrial alterations in human cardiomyopathy. Images retrieved from the publicly available dataset by [108]. (**a**) Heatmap illustrating the normalized expression levels of mitochondrially encoded genes in cardiac muscle cells, averaged across individual samples. (**b**) Violin plots showing the distribution of normalized mitochondrial gene expression across different disease groups. The violin plots are centered around the mean value with standard deviation, and the width of each plot reflects the distribution of sample values. *p*-values were determined using the Mann–Whitney U test; only significant *p*-values are shown (* *p* < 0.05, ** *p* < 0.01, *** *p* < 0.001). Abbreviations: NF, non-failing hearts; DCM, dilated cardiomyopathy; HCM, hypertrophic cardiomyopathy.

**Table 1 cells-14-00778-t001:** Main studies on the cGAS/STING pathway in cardiomyopathy. *In vitro* models are indicated in italics in the table.

Cardiomyopathy Type	Model (In Vivo–*In Vitro*)	Key Findings	Ref
Sepsis-induced cardiomyopathy (SIC)	LPS-injected mice; *H9C2 cells, neonatal rat cardiomyocytes (NRCMs)*	STING knockdown suppresses IRF3 activation, reduces inflammation, apoptosis, and pyroptosis	[24]
	Human blood; LPS-injected mice; *RAW264.7 macrophages; H9C2 cells*	Islet cell autoantigen of 69 kDa (ICA69) deletion inhibits STING-mediated inflammation and ferroptosis	[25]
	LPS-injected mice; *H9C2 cells*	cGAS knockdown or ALDH2 treatment reduces STING pathway activation	[26]
Dilated cardiomyopathy (DCM)	*Myh6-Cre:Lmna*^F/F^: *Crat*^-/-^ mice; *Neonatal rat ventricular myocytes (NRVMs)*	cGAS knockdown reduces IFN-stimulated gene expression	[27]
	Human DCM hearts	Elevated cGAS in human primary DCM samples; STING unchanged	[28]
	LMNA-DCM mice (with/without cGAS)	cGAS deletion improves survival and cardiac function	[29]
	LMNA-DCM mice: (*Myh6-MerCreMer:Lmna^F^*^/F^)	No activation of cGAS/STING in cardiomyocytes, cGAS or STING knockout does not rescue the phenotypes of LMNA-DCM	[30]
Diabetic cardiomyopathy	Diabetic (db/db) mice; *Palmitic acid (PA)-treated H9C2 cells (rat cardiomyocytes)*	Mitochondrial mtDNA activates cGAS/STING; STING inhibition in H9C2 cardiomyocytes (by C176) reduces inflammation and apoptosis	[31]
	*PA-treated H9C2 cells*	Cytosolic mtDNA activates cGAS/STING; knockdown inhibits pyroptosis	[32]
	Streptozotocin (STZ)-treated (db/db) mice; *NRCMs*	Meteorin-like hormone (Metrnl) inhibits cGAS/STING in cardiomyocytes and activates the autophagy pathway	[33]
	STZ-treated and high-fat-diet (HFD)-fed mice; *NRCMs*	BRG1 loss activates STING, worsening inflammation and apoptosis induced by hyperglycemia and hyperlipidemia	[34]
	STZ-treated and HFD-fed mice; human blood	Fibroblasts engulf mtDNA vesicles, activating cGAS/STING	[35]
	STZ-treated and HFD-fed mice; *H9C2 cells*	Irisin and mitochondrial ubiquitin ligase (MITOL) inhibit cGAS/STING, improving cardiac function	[36]
Other Types: TMEM43 arrhythmogenic cardiomyopathy	*Tmem43* mutant mice: *Myh6-Cre*: *Tmem43*^W/F^ mice	STING activated at later stages; related to DNA damage signals	[37]
Chagas cardiomyopathy	T. cruzi-infected mice; *Murine bone marrow cells, macrophages*	cGAS/STING senses T. cruzi vesicles, promotes inflammation	[38]
Stress cardiomyopathy	Ovariectomized mice treated with isoproterenol; *RAW264.7 macrophages*	Ginsenoside Rb1 suppresses STING-mediated macrophage inflammation	[39]
LEMD2 arrhythmogenic cardiomyopathy	*Lemd2* mutant mice (*Lemd2 p.L13R* knock-in); *HeLa LEMD2 p.L13R KI cells*	Nuclear envelope rupture recruits cGAS, activates STING/IFN signaling	[40]
Doxorubicin-induced cardiomyopathy	Doxorubicin-treated mice (acute injury)	STING knockdown reduces vacuolization and myofibril loss and improves function	[41]
	Low-dose Doxorubicin-treated mice (chronic injury); *human cardiac microvascular endothelial cells (HCMECs)*	Global and endothelial-cell-specific STING deletion ameliorates cardiotoxicity and endothelial dysfunction	[21]

**Table 2 cells-14-00778-t002:** Molecular interventions in the cGAS/STING pathway in cardiomyopathy. ↑ indicates upregulation or increased occurrence, ↓ indicates downregulation or reduced occurrence of the metabolites or signaling cascades listed in the table.

Target	Compound/Drug	Mode of Action	Effects on Signaling Cascades and in Animal Models	Ref
cGAS	RU-compounds (RU.365, RU.521)	Catalytic site inhibitor	Reduced expression levels of *Ifnb1* mRNA in *Trex* knockout mice (which constitutively activate cGAS) ↓ IL-1β, ↓ cleaved caspase-3 ↓ Apoptosis	[34]
	Antimalarial drugs (i.e., Hydroxychloroquine, Quinacrine)	Disrupting dsDNA binding	Hydroxychloroquine and Quinacrine inhibit dsDNA binding to cGAS In vitro: ↓ IFN-β expression In vivo: ↓ Early IFN-1 response in Hydroxycloroquine-treated mice	[101,102]
	Suramin	Disrupting dsDNA binding	Suramin inhibits dsDNA binding to cGAS in vitro (THP1-Dual cells) ↓ IFN-β expression (mRNA and protein)	[103]
	Acetylsalicylic acid	cGAS acetylation and inhibition	↓ IFN-production in vitro (THP-1 cells) ↓ Expression of interferon-stimulated genes (ISG) Trex1^–/–^ bone marrow cells ↓ ISG expression in the hearts of Trex1^–/–^ mice	[104]
STING and TBK1	Astin C	STING inhibition—targeting the cyclic dinucleotide binding site	↓ Expression of Ifnb, Cxcl10, Isg15, Isg56 and Tnf mRNA in the heart of Trex1^-/-^ mice (in vivo) ↓ Expression of type 1 interferone in Trex1^-/-^ Bone marrow cells (in vitro)	[105]
	Nitrofuran derivatives - C176 and C178	STING inhibition—covalent binding to cysteine residue 91, inhibiting palmitoylation and activation of STING	↓ Serum levels of type I interferons and IL-6 in Trex1^−/−^ mice	[106]
			↓ Phosphorylation of p65 ↑ Improves diastolic cardiac function ↑ Partially improve myocardial hypertrophy	[31]
			↓ Cardiac IRF3 phosphorylation, IRF3 nuclear translocation, and CD38 expression ↑ Cardiomyocyte NAD levels, mitochondrial function, and ↑ left ventricular systolic function ↓ Cardiomyocyte apoptosis ↓ Antitumor effects of doxorubicin	[21]
			↓ IL-1β, cleaved caspase-3 No effect on γ-H2AX ↓ Apoptosis	[34]
	Amlexanox	TBK1 inhibitor	Same effect as C176	[21]
	3-acylaminoindole derivative - H-151	STING inhibition—blocking the activation-induced palmitoylation and clustering of STING	↓ Calf thymus DNA-induced production of TNF in a dose-dependent manner	[39]
			↓ Reduces IFN-β levels in a dose-dependent manner	[107]
	Ginsenoside Rb1	Major chemical constituent of ginseng; suppressing the activation of STING	↓ STING-mediated proinflammatory activation of macrophages ↓ Myocardial fibrosis and inflammatory responses in the heart ↓ DNA-triggered proinflammatory activation of macrophages ↓ DNA-triggered whole-genome gene expression alterations in macrophages	[39]
	DMXAA	STING agonist	↑ STING phosphorylation. ↑ TNF, IL6, CCL2, IFN-β	[39]

## Data Availability

No new data were created or analyzed in this study.

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
