# Peer review of "The cGAS/STING Pathway: Friend or Foe in Regulating Cardiomyopathy"

_cells, 2025, doi:10.3390/cells14110778_

Round 1
Reviewer 1 Report
Comments and Suggestions for Authors
This is a well-written and focused review that effectively summarizes the role of the cyclic guanosine monophosphate (GMP)-adenosine monophosphate (AMP) synthase/stimulator of interferon genes (cGAS/STING) pathway in cardiomyopathy, compiling and critically analyzing the most relevant experimental evidence published to date.
The authors emphasize the cGAS/STING signaling pathway, which plays a central role in the innate immune response by detecting cytoplasmic DNA and triggering inflammation and apoptosis. Given the key role of inflammation in the development and progression of various forms of cardiomyopathy, the involvement of cGAS/STING signaling appears to be causative and of high translational interest.
The review’s main objective is to highlight this specific signaling pathway in the context of cellular damage and to explore new therapeutic targets for a disease that currently lacks curative treatments. This aim is clearly presented and well justified. The discussion of the different cardiomyopathies in which the cGAS/STING axis has been implicated is thorough, and the literature is critically and clearly reviewed.
Minor suggested revision:
I would recommend revising Table I. In its current form, the table contains an excess of information, which makes it difficult to consult. A more selective and schematic presentation of key results would significantly improve its clarity and utility for the reader.
Author Response
Reviewer #1
Q1: I would recommend revising Table I. In its current form, the table contains an excess of information, which makes it difficult to consult. A more selective and schematic presentation of key results would significantly improve its clarity and utility for the reader.
Response: Thank you for the comment. We have modified the table accordingly. The revised tables are as follows:
Table 1. Main studies on the cGAS/STING pathway in cardiomyopathy
Cardiomyopathy Type |
Year |
Model (in vivo and/or in vitro) |
Key Findings |
Ref |
Sepsis-Induced cardiomyopathy (SIC) |
2019 |
LPS-injected mice; H9C2 cells, Neonatal rat cardiomyocytes (NRCMs) |
STING knockdown suppresses IRF3 activation, reduces inflammation, apoptosis and pyroptosis |
[24] |
2022 |
Human blood; LPS-injected mice; RAW264.7 macrophages; H9C2 cells |
slet cell autoantigen of 69 kDa (ICA69) deletion inhibits STING-mediated inflammation and ferroptosis |
[25] |
|
2023 |
LPS-injected mice; H9C2 cells |
cGAS knockdown or ALDH2 treatment reduces STING pathway activation |
[26] |
|
Dilated cardiomyopathy (DCM) |
2023 |
Myh6-Cre:LmnaF/F:Crat-/- mice; Neonatal rat ventricular myocytes (NRVMs) |
cGAS knockdown reduces IFN-stimulated gene expression |
[27] |
2023 |
Human DCM hearts |
Elevated cGAS in human primary DCM samples ; STING unchanged |
[28] |
|
2022 |
LMNA-DCM mice (with/without cGAS) |
cGAS deletion improves survival and cardiac function |
[29] |
|
2024 |
LMNA-DCM mice: (Myh6-MerCreMer:LmnaF/F) |
No activation of cGAS/STING in cardiomyocytes, cGAS or STING knockout does not rescue the phenotypes of LMNA-DCM |
[30] |
|
Diabetic cardiomyopathy |
2022 |
diabetic (db/db) mice; Palmitic acid (PA)-treated H9C2 cells (rat cardiomyocytes) |
Mitochondrial mtDNA activates cGAS/STING; STING inhibition in H9C2 cardiomyocytes (by C176) reduces inflammation and apoptosis |
[31] |
2022 |
PA-treated H9C2 cells |
Cytosolic mtDNA activates cGAS/ STING; knockdown inhibits pyroptosis |
[32] |
|
2023 |
streptozotocin (STZ) -treated (db/db) mice; NRCMs |
Meteorin-like hormone (Metrnl) inhibits cGAS/STING in cardiomyocytes and activates the autophagy pathway |
[33] |
|
2023 |
STZ-treated and high-fat diet (HFD) fed mice; NRCMs |
BRG1 loss activates STING, worsening inflammation and apoptosis induced by hyperglycemia and hyperlipidemia |
[34] |
|
2024 |
STZ-treated and HFD fed mice; human blood |
Fibroblasts engulf mtDNA vesicles, activating cGAS/STING |
[35] |
|
2024 |
STZ-treated and HFD fed mice; H9C2 cells |
Irisin and mitochondrial ubiquitin ligase (MITOL) inhibit cGAS/STING, improving cardiac function |
[36] |
|
Other Types: TMEM43 arrhythmogenic cardiomyopathy
|
2021 |
TMEM43 mutant mice: Myh6-Cre:Tmem43W/F mice |
STING activated at later stages; related to DNA damage signals |
[37] |
Chagas cardiomyopathy |
2020 |
T. cruzi-infected mice; Murine bone marrow cells, macrophages |
cGAS/STING senses T. cruzi vesicles, promotes inflammation |
[38] |
Stress cardiomyopathy |
2024 |
Ovariectomized mice treated with isoproterenol; RAW264.7 macrophages |
Ginsenoside Rb1 suppresses STING-mediated macrophage inflammation |
[39] |
LEMD2 arrhythmogenic cardiomyopathy |
2023 |
LEMD2 mutant mice (Lemd2 p.L13R knock-in); HeLa LEMD2 p.L13R KI cells |
Nuclear envelope rupture recruits cGAS, activates STING/IFN signaling |
[40] |
Doxorubicin-induced cardiomyopathy |
2023 |
Doxorubicin treated mice (acute injury) |
STING knockdown reduces vacuolization and myofibril loss and improves function |
[41] |
2023 |
low-dose Doxorubicin treated mice (chronic injury); human cardiac microvascular endothelial cells (HCMECs) |
Global and endothelial cell-specific STING deletion ameliorates cardiotoxicity and endothelial dysfunction |
[21] |
Table 2. Molecular interventions in the cGAS/STING pathway in cardiomyopathy
Target |
Compound / Drug |
Mode of action |
Effects on signaling cascades and in animal models |
Ref |
cGAS |
RU-compounds (RU.365, RU.521) |
catalytic site inhibitor |
reduced expression levels of Ifnb1 mRNA in Trex knockout mice (which constitutively activate cGAS) ∙ ↓ IL-1β, ↓ cleaved caspase-3 ∙ ↓ apoptosis |
[34] |
Antimalarial drugs (i.e. Hydroxychloroquine, Quinacrine) |
disrupting dsDNA binding |
Hydroxychloroquine and Quinacrine inhibit dsDNA binding to cGAS in vitro: ↓ IFN-β expression In vivo: ↓ early IFN-1 response in Hydroxycloroquine-treated mice |
[101,102] |
|
Suramin |
disrupting dsDNA binding |
suramin inhibits dsDNA binding to cGAS in vitro THP1-Dual cells: ↓ IFN-β expression (mRNA and protein) |
[103] |
|
Acetylsalicylic acid |
cGAS acetylation and inhibition |
↓ IFN-production in vitro (THP-1 cells) and ↓ expression of interferon-stimulated genes (ISG) Trex1–/– bone marrow cells; ↓ ISG expression in the hearts of Trex1–/– mice |
[104] |
|
STING and TBK1 |
Astin C |
STING inhibition - targeting the cyclic dinucleotide binding site |
↓ expression of Ifnb, Cxcl10, Isg15, Isg56 and Tnf mRNA in the heart of Trex1-/- mice (in vivo); ↓ expression of type 1 interferone in Trex1-/- bone marrow cells (in vitro) |
[105] |
Nitrofuran derivatives
- C176 and C178 |
STING inhibition - Covalent binding to cysteine residue 91, inhibiting palmitoylation and activation of STING |
↓ serum levels of type I interferons and IL-6 in Trex1−/− mice |
[106] |
|
↓ phosphorylation of p65 ↑ improve diastolic cardiac function ↑ Partially improve myocardial hypertrophy |
[31] |
|||
↓ cardiac IRF3 phosphorylation, IRF3 nuclear translocation and CD38 expression. ↑ cardiomyocyte NAD levels, mitochondrial function and ↑ left ventricular systolic function. ↓ cardiomyocyte apoptosis. ↓ antitumor effects of doxorubicin |
[21] |
|||
∙ ↓ IL-1β, cleaved caspase-3; ∙ no effect on γ-H2AX; ∙ ↓ apoptosis |
[34] |
|||
Amlexanox |
TBK1 inhibitor |
Same effect as C176 |
[21] |
|
3-acylaminoindole derivative
- H-151 |
STING inhibition - blocking the activation-induced palmitoylation and clustering of STING |
↓ calf thymus DNA-induced production of TNF in a dose-dependent manner |
[39] |
|
↓ reduced IFN-β levels in a dose-dependent manner |
[107] |
|||
Ginsenoside Rb1 |
major chemical constituent of ginseng; suppressing the activation of STING |
↓ STING-mediated proinflammatory activation of macrophages. ↓ myocardial fibrosis and inflammatory responses in the heart. ↓ DNA-triggered proinflammatory activation of macrophages. ↓ DNA-triggered whole-genome gene expression alterations in macrophages; |
[39] |
|
DMXAA |
STING agonist |
↑ STING phosphorylation. ↑ TNF, IL6, CCL2, IFN-β; |
[39] |
Reviewer 2 Report
Comments and Suggestions for Authors
This is a fine study investigating about the impact of cGAS/STING pathway in the development of cardiomyopathy. First, the authors summarized the key findings on the cGAS/STING pathway in different cardiomyopathies, using in vivo/in vitro models as well as clinical samples. Next, they considered about the impact of the intervention against cGAS/STING pathway. Finally, they investigated about existing molecular interventions and highlighted potential therapeutic targets to mitigate cGAS/STING-driven inflammation. The review was finely arranged, however there were several issues to be addressed.
# The authors picked up some kinds of cardiomyopathy, however myocarditis is one of the most relevant etiology of cardiomyopathy according to the immune imbalance.
# Table 1 and 2 were much redundant. These should be summarized more concisely.
# They picked up two genetic cardiomyopathy (LMNA and LEMD), however there are many other kinds of genetic cardiomyopathy. How about the impact of cGAS/STING in other genetic cardiomyopathy?
There were several typos.
L30 by it agonists and antagonists →by its? agonist....
L139 cardiac fibrosis and apoptosis39?
Author Response
We thank the reviewer for the constructive suggestions. We have revised the manuscript and please see the attachment of point-to-point response.

Reviewer 3 Report
Comments and Suggestions for Authors
The manuscript by Wang and co-authors is a comprehensive review on the impact of the cGAS/STING pathway in various types of human cardiomyopathy, along with molecular interventions targeting this pathway as therapeutic approaches towards alleviating inflammatory cardiomyopathy. This is an interesting and well-written review, providing clear descriptions of experimental evidence on the contribution of cGAS/STING in the different cardiomyopathies. In addition, the authors provide possible explanations in cases of contradictory findings.
Comments:
1) The meta-analysis of human single-cell RNA-seq data is an interesting and important addition to the manuscript. Based on the findings obtained from the analysis of the single-study by Chaffin et al, decreases in particular mitochondrial gene expression were identified in DCM and HCM vs control non-failing patients. The authors mention that mitochondrial DNA is significantly reduced in cardiac muscle (line 386) of the patients, however this is slightly misleading as the RNA-seq data indicate changes in expression levels rather that DNA content itself. Is there any information regarding inflammation in these patients? Have the authors checked for expression changes in the cGAS/STING pathway?
2) Why did the authors choose to analyze specifically the study by Chaffin et al? Are there bioinformatical data available from other cardiomyopathy patient categories that may corroborate these mitochondrial changes findings? These may not necessarily be restricted to single-cell studies but to cardiac gene expression studies.
3) Mitochondria are already known to be closely linked to cardiac disease, so is there any information regarding the altered genes identified by the authors and their protein expression level, localization and/or function in cardiac disease?
4) Section 4 is very brief and consists primarily of Table 2. Given that this is an important section on interventions and therapeutic approaches targeting cGAS/STING pathway in cardiomyopathy, the authors would be advised to expand by briefly discussing the key findings on some of the most promising compounds/drugs.
5) Along the same lines, as multiple molecular mechanisms have been discussed in the manuscript, what do the authors believe is the most promising molecular pathway/mechanism to target? The authors mention mitochondria (line 396) but it is unclear whether this is based on the bioinformatical data of section 4.2 or on experimental evidence.
Minor comment:
1) Please check the format of in-text citation of references, especially for ref 39 (line 134), 80 (line 299) and 103 (line 428).
2) I assume that C- caspase3 and T-caspase3 (line 283) means cleaved and total but please define.
Author Response
We thank the reviewer for the fruitful suggestions. We have revised the manuscript accordingly and attached please find the point-to-point response.

Round 2
Reviewer 3 Report
Comments and Suggestions for Authors
Thank you for addressing my questions and suggestions. I have no further comments.
Author Response
We thank the reviewer for the constructive suggestions!